# Transcriptomic Analysis of Tobacco Plants in Response to Whitefly Infection

**DOI:** 10.3390/genes14081640

**Published:** 2023-08-18

**Authors:** Xin Wang, Zhuang-Xin Ye, Yi-Zhe Wang, Xiao-Jing Wang, Jian-Ping Chen, Hai-Jian Huang

**Affiliations:** 1College of Life Sciences, Fujian Agriculture and Forestry University, Fuzhou 350002, China; wangzhichaowork@163.com; 2State Key Laboratory for Managing Biotic and Chemical Threats to the Quality and Safety of Agro-Products, Key Laboratory of Biotechnology in Plant Protection of Ministry of Agriculture and Zhejiang Province, Institute of Plant Virology, Ningbo University, Ningbo 315211, China; yzx244522794@163.com (Z.-X.Y.); 2011074048@nbu.edu.cn (Y.-Z.W.); 2211130055@nbu.edu.cn (X.-J.W.)

**Keywords:** transcriptome, plant defense, *Bemisia tabaci*, *Nicotiana tabacum*

## Abstract

The whitefly *Bemisia tabaci* is one of the most destructive pests worldwide, and causes tremendous economic losses. Tobacco *Nicotiana tabacum* serves as a model organism for studying fundamental biological processes and is severely damaged by whiteflies. Hitherto, our knowledge of how tobacco perceives and defends itself against whiteflies has been scare. In this study, we analyze the gene expression patterns of tobacco in response to whitefly infestation. A total of 244 and 2417 differentially expressed genes (DEGs) were identified at 12 h and 24 h post whitefly infestation, respectively. Enrichment analysis demonstrates that whitefly infestation activates plant defense at both time points, with genes involved in plant pattern recognition, transcription factors, and hormonal regulation significantly upregulated. Notably, defense genes are more intensely upregulated at 24 h post infestation than at 12 h, indicating an increased immunity induced by whitefly infestation. In contrast, genes associated with energy metabolism, carbohydrate metabolism, ribosomes, and photosynthesis are suppressed, suggesting impaired plant development. Taken together, our study provides comprehensive insights into how plants respond to phloem-feeding insects, and offers a theoretical basis for better research on plant–insect interactions.

## 1. Introduction

Plants and insects have engaged in a coevolutionary relationship spanning hundreds of millions of years, leading to the development of diverse strategies for combatting each other [1]. In the face of herbivorous insect attacks, plants have evolved various defense mechanisms. On the one hand, they produce specialized physical barriers, secondary metabolites, and toxic compounds to counteract herbivore assaults [2]. On the other hand, plants have developed inducible defenses, which can be categorized as direct or indirect defense strategies [3]. Indirect defense involves the production and emission of herbivore-induced plant volatiles (HIPVs), which serve as crucial cues for parasitoids and predators, ultimately reducing damage inflicted on the plant [4]. Conversely, direct defense mechanisms encompass physical barriers like leaf surface wax, thorns, and trichomes, constituting the first line of defense against herbivores [1]. Additionally, plants activate specific genes associated with the synthesis of secondary metabolites. Noteworthy compounds such as gossypol, glucosinolates, alkaloids, phenolics, and proteinase inhibitors have been identified to impede the development of herbivores [5,6].

The plant defense system is intricately regulated by sophisticated signaling networks, which involve phytohormone signaling pathways, reactive oxygen species (ROS) generation, transcription factors (TFs), kinase cascades, and mitogen-activated protein kinases (MAPKs) [7]. Pattern recognition receptors (PRRs) serve as the first line of defense, capable of recognizing conserved microbe-associated molecular patterns (MAMPs) and damage-associated molecular patterns (DAMPs), thereby activating primary innate immunity [8,9,10]. These receptors, located on the surface of plant cells, comprise receptor-like kinases (RLKs) and receptor-like proteins (RLPs), which can sense environmental changes, and transmit this information via activated signaling pathways to trigger adaptive responses [11,12]. In addition to PRRs, TFs play a crucial role in plant defense systems, including WRKY TFs, zinc finger TFs, MYB TFs, and ethylene-responsive TFs (ERFs). Previous research has demonstrated the involvement of TFs in plant–insect interactions, regulating herbivore-induced plant defense responses. For instance, in *Arabidopsis thaliana*, AtMYC2 was identified as a regulator of JA-mediated plant herbivore resistance [13], while OsWRKY53 activates rice defenses against brown planthoppers by activating an H_2_O_2_ burst and suppressing ethylene biosynthesis [14].

The whitefly *B. tabaci* (Gennadius) (Hemiptera: Aleyrodidae) is a prominent piercing–sucking herbivore known for causing significant crop losses through direct feeding and transmitting viruses [15,16]. As a highly polyphagous species, *B. tabaci* exhibits a wide host range, including ornamentals and greenhouse crops such as tomato, pepper, beans, eggplant, cucumber, and tobacco [17]. Among these crops, tobacco (*N. tabacum*), which serves as a major non-food species and a model organism, produces an array of plant defense metabolites that are toxic to most herbivorous insects. However, the mechanisms by which whiteflies manipulate plant defenses to adapt to tobacco plants, as well as the tobacco response to whitefly infestation, remain unclear. The advent of transcriptome technologies has provided an ideal method for studying plant physiological processes in response to *B. tabaci*. In this study, we conducted deep RNA sequencing of tobacco plants at different time points following whitefly infestation, unveiling the complex interaction between whiteflies and their tobacco hosts.

## 2. Materials and Methods

### 2.1. Plants and Whitefly

We utilized tobacco plants of the *N. tabacum* cv. NC89 variety, which were cultivated in climate chambers at a temperature of 24 ± 2 °C, under a 14 h light and 10 h dark cycle, and with 50–70% humidity. The seeds for these plants were provided by Li-Rong Pan from the Institute of Biotechnology, Zhejiang University.

The Mediterranean (MED) species of the *B. tabaci* complex were used in this study. The insects were established from a field whitefly sample collected in Suzhou, China, in 2019. The whitefly colony was maintained on tobacco plants of the *N. tabacum* cv. NC89 variety. The whiteflies were reared in cages placed inside a climate chamber at a temperature of 27 ± 2 °C, under a 14 h light and 10 h dark cycle, and with 50–70% humidity.

### 2.2. Plant Treatments

To investigate the impact of whitefly feeding on tobacco plants, we subjected the plants to different treatments. Briefly, the tobacco plants were transferred to a climate chamber (27 ± 2 °C) one day before treatments. MED1 whitefly adults were released onto tobacco plants at the 4–5 leaf stage. The whiteflies were allowed to feed on the plants for 12 h and 24 h, respectively, under the same growth conditions. Non-infested tobacco plants were used as controls. Three biological replicates were performed.

### 2.3. RNA Isolation and Illumina Sequencing

Total RNA was isolated from the plant samples using a TRIzol Total RNA Isolation Kit (Takara, Dalian, China), according to the manufacturer’s instructions. Three biological replicates were performed for each treatment.

The extracted RNA samples were sent to Novogene (Tianjin, China) for RNA sequencing. RNA integrity was assessed using the Fragment Analyzer 5400 (Agilent Technologies, Palo Alto, CA, USA). After quantifying the total RNA, nine cDNA libraries were constructed according to the manufacturer’s recommendations of NEBNext^®^ UltraTM RNA Library Prep Kit for Illumina^®^ (NEB, Boston, MA, USA) and index codes were added to attribute sequences to each sample. Briefly, the mRNA was isolated from the total RNA using poly-T oligo-attached magnetic beads. The fragmentation of the mRNA was performed under elevated temperatures using divalent cations in NEBNext First Strand Synthesis Reaction Buffer (5 × X). The first strand cDNA synthesis was carried out using a random hexamer primer and M-MuLV Reverse Transcriptase. Subsequently, the second strand cDNA synthesis was performed using DNA Polymerase I and RNase H. The remaining overhangs were then converted into blunt ends using exonuclease/polymerase activities. Following the adenylation of the 3’ ends of DNA fragments, NEBNext Adaptors with hairpin loop structures were ligated to prepare the fragments for hybridization. To select cDNA fragments with a preferred length of 250–300 bp, the library fragments were purified using the AMPure XP system (Beckman Coulter, Beverly, MA, USA). Subsequently, 3 µL of USER Enzyme (NEB, Ipswich, MA, USA) was applied to the size-selected, adaptor-ligated cDNA and incubated at 37 °C for 15 min, followed by 5 min at 95 °C, prior to PCR amplification. PCR was performed using Phusion High-Fidelity DNA polymerase, Universal PCR primers, and Index (X) Primer. Finally, the PCR products were purified using the AMPure XP system, and the quality of the libraries was assessed using the Agilent Bioanalyzer 2100 system.

The index-coded samples were clustered using the TruSeq PE Cluster Kit v3-cBot-HS (Illumina, San Diego, CA, USA) on a cBot Cluster Generation System, following the manufacturer’s instructions. The sequencing was performed using the Illumina Novaseq 6000 platform with 150-bp paired-end reads. For each sample, cDNA library construction and Illumina sequencing were carried out with three technical replicates. The output data were submitted to the National Genomics Data Center under accession number PRJCA018862.

The internal software Fastp version 0.23.1 [18] was used to obtain clean reads by removing low-quality reads that contained adapters, empty reads or reads with unknown sequences “N” from the raw data. The quality control of reads was assessed using the FastQC program (version 0.11.3) [19]. This program executed a series of analysis modules on raw data and generated a report with statistics for the data analyzed, including the error rate of average base sequencing and the percentage of bases with Phred quality scores > 20 (Q20) or >30 (Q30). Subsequently, the clean reads from each cDNA library were aligned to the reference genome sequences of *N. tabacum* (https://solgenomics.net/ftp/genomes/Nicotiana_tabacum/edwards_et_al_2017/assembly/, accessed on 1 September 2022) using Hierarchical Indexing for Spliced Alignment of Transcripts (HISAT2) [20]. The low-quality alignments were filtered with Sequence Alignment/Map tools (SAMtools) [21].

### 2.4. DEGs Expression Analysis

Transcripts per million (TPM) expression values were calculated using cufflink [22]. The DESeq2 (http://bioconductor.org/packages/release/bioc/html/DESeq2) was used for identifying the differentially expressed genes with default parameters. Based on these statistical analyses, genes meeting the criteria of *p*-value < 0.01 and an absolute value of the log_2_ ratio > 1 were deemed significant differentially expressed genes (DEGs). To gain insights into the potential functions and metabolic pathways of the DEGs, further analysis was conducted for GO and KEGG enrichments using TBtools software v1.0697 [23]. In this software, enriched *p*-values were calculated according to one-sided hypergeometric test P=1−∑i=0m−1MiN−Mn−iNn, with *N* representing the number of the gene with KEGG/GO annotation, *n* representing the number of DEGs in *N*, *M* representing the number of genes in each KEGG/GO term, and *m* representing the number of DEGs in each KEGG/GO term. The significant pathways were defined based on a corrected *p*-value ≤ 0.05. PCA was used to reveal overall differences in gene expression patterns among different transcriptomes, and R function plotPCA (github.com/franco-ye/TestRepository/blob/main/PCA_by_deseq2.R, accessed on 1 September 2022) was also used.

### 2.5. Quantitative Real-Time PCR (qRT-PCR)

Candidate transcriptomic genes were validated with qRT-PCR analysis using three biological replicates. Total RNA was extracted as described above. cDNAs were synthesized using the HiScript II Reverse Transcriptase (Vazyme, Nanjing, China). qRT-PCR was performed on a Roche Light Cycler^®^ 480 Real-Time PCR System (Roche Diagnostics, Mannheim, Germany) using the Hieff qPCR SYBR Green Master Mix (Low Rox Plus) (Yeasen Biotechnology, Shanghai, China). The PCR procedure was as follows: denaturation for 5 min at 95 °C, followed by 40 cycles at 95 °C for 10 s and 60 °C for 30 s. The primers used in qRT-PCR were designed using Primer Premier v6.0 (Appendix A). The reference gene Tubulin was used for normalization. The relative expression levels of the target genes were calculated using the 2^−∆∆Ct^ method.

## 3. Results

### 3.1. Transcriptomic Profile of Tobacco Plants Infested with Whiteflies

To evaluate the global transcriptomic profile of tobacco in response to whitefly infestation, we performed deep RNA-Seq of the tobacco NC89 cultivars following whitefly infestation for 12 and 24 h (hereafter referred to as I-12 h and I-24 h). The non-infested tobacco plants were used as controls (hereafter referred to as NI), and three biological replicates were conducted for each treatment. In total, we generated approximately 2.3 hundred million paired-end (PE) reads from these nine libraries, with approximately 23–28 million reads generated per library. The raw reads underwent sequence duplication and GC content analysis using FastQC software (Table 1). The summaries of the total clean/raw reads, the total number of bases in the clean/raw data, and the error rate of average base sequencing, Q20, and Q30 are presented in Table 1. Additionally, principal component analysis (PCA) revealed significant changes in the tobacco transcriptomic profile induced by whitefly infestation. PC1 and PC2 explained 91% and 4% variance, respectively, and the samples from the three biological replicates exhibited good reproducibility (Appendix A).

### 3.2. Differently Expressed Genes (DEGs) between Whitefly-Infested and Non-Infested Tobacco

To evaluate alterations in the gene expression in tobacco plants in response to infestation by *B. tabaci*, we conducted a comprehensive analysis of DEGs. We performed a comparative analysis of RNA libraries from non-infested tobacco plants and those infested with whiteflies for 12 and 24 h, resulting in two distinct categories: (1) early response, a comparison between the control group and tobacco plants infested for 12 h, and (2) late response, a comparison between the control group and tobacco plants infested for 24 h.

For the early response, a total of 244 genes were differentially expressed, including 128 upregulated genes and 116 downregulated genes (Appendix A, Figure 1a). For the late response, there were 2417 DEGs, with 983 upregulated and 1434 downregulated (Appendix A, Figure 1a). To gain insights into the functions of these differentially expressed genes, we performed gene ontology (GO) enrichment analysis for both types of DEGs. We found that the upregulated genes in both the early and late stages were primarily associated with plant defense pathways, such as response to stimuli, response to biotic stimuli, response to hormones, response to reactive oxygen species, and innate immune responses (Figure 2a,b). These results suggest that whitefly infestation activates the plant innate immune system, particularly the pathways involving reactive oxygen species (ROS) and hormones, which are crucial components of the plant’s sophisticated signaling networks. In contrast, the downregulated genes in both early and late stages were mainly associated with cellular components of plant organelles, such as cytoplasm, chloroplast, and thylakoid (Figure 2c,d). Several genes related to photosynthesis were also downregulated. Previous studies have highlighted the critical role of plant organelles, such as chloroplasts and cytoplasm, in plant immunity [24,25]. The downregulated expression of organelle-related genes suggests that *B. tabaci* potentially inhibits organelle-mediated immunity to facilitate feeding. There were 89 DEGs specifically differentially expressed at 12 h, but not 24 h, when compared with the non-infested control. Notably, the majority of these genes (68, 76.4%) were specifically upregulated at the early stage, including ABC transporters, serine threonine tyrosine kinase, calmodulin-binding protein, BHLH transcription factor, and UDP-glucosyltransferase. These genes potentially act as immediate-early expressed genes [26] that efficiently respond to whitefly infestation.

### 3.3. Plant Signal Networks Response to Whitefly Attacks

To gain a better understanding of the plant immune response triggered by *B. tabaci* infestation and the key resistance genes involved in this process, we conducted a comparative analysis of DEGs between the two post-whitefly infestation time points (12 and 24 h). We found that a total of 54 genes were upregulated and 91 genes were downregulated (Figure 1b). Among the upregulated genes, the WRKY TF1 exhibited the highest upregulation among all upregulated genes. Additionally, other WRKY TFs were upregulated in both early and late stages. Specifically, in the late stage, WRKY TFs, including WRKY1, WRKY2, WRKY3, WRKY5, WRKY6, WRKY35, and WRKY78, were upregulated (Figure 3b). These findings indicate that WRKY genes play important roles in tobacco resistance against whitefly attacks. Interestingly, we also observed the upregulation of five ERFs in response to whitefly infestation, including ERF1, ERF1a, ERF2b, ERF4, and ERF5 (Figure 3c).

In addition to TFs, a significant number of genes associated with PRRs exhibited notable changes. PRRs consist of RLKs and RLPs and are involved in specifically recognizing herbivory-associated cues, such as oral secretions and eggs, thus activating plant immune responses [27,28]. We found that 49 PRR genes were altered, including 26 upregulated and 23 downregulated genes, upon whitefly infestation (Figure 3a). Overall, these findings suggest the activation of TFs and cell surface receptor protein kinases involved in regulating plant immunity in response to whitefly infestation.

### 3.4. Metabolic Processes Affected by Whitefly Attack

To investigate how *B. tabaci* regulates plant defenses, we examined DEGs in relation to metabolic pathways, particularly focusing on downregulated DEGs. The results of the Kyoto Encyclopedia of Genes and Genomes (KEGG) pathway analysis revealed that the downregulated DEGs were primarily associated with metabolic pathways. The commonly downregulated DEGs at both early and late stages were enriched in pathways such as metabolism, energy metabolism, ribosomes, and carbohydrate metabolism (Figure 4a). Furthermore, whitefly infestation resulted in enrichment in other metabolism-related pathways, particularly at a late stage (Figure 4b,c). These results indicate that feeding by *B. tabaci* can impact fundamental metabolic and energy processes in plants. Metabolic pathways are essential for plant growth, development, and defense against biotic and abiotic stresses. The downregulation of genes associated with metabolism suggests that *B. tabaci* may inhibit basal metabolism and defense-related metabolic compounds as a strategy to overcome plant defenses.

### 3.5. Validation of DEGs Using Quantitative Real-Time PCR (qRT-PCR)

To confirm the expression of the DEGs, 10 genes that were associated with plant defenses or showed dramatic upregulation/downregulation in the RNA-Seq analysis were selected for qRT-PCR analysis (Appendix A). The validation was conducted using three independent biological replicates. We observed the upregulation of zinc finger proteins and ATP-binding cassette (ABC) transporters. In contrast, two MYB TFs were downregulated. MYB TFs are a large protein family involved in regulating plant growth, development, and stress adaptation. Furthermore, peroxidase 40, L-ascorbate peroxidase and cryptochrome 1b, which are related to plant defense, were downregulated. Additionally, rubisco activate 1 was upregulated, while two late embryogenesis abundant protein 5s were downregulated. Overall, the expression profile of the genes selected for qRT-PCR validation was consistent with the DEG data (Figure 5), which indicates that the RNA-Seq data are reliable and reproducible.

## 4. Discussion

Whiteflies are among the most economically important pests of crops worldwide, causing significant losses in agricultural productivity each year. Despite previous studies on the whitefly–plant interaction, the ongoing evolutionary arms race between plants and whiteflies necessitates a deeper understanding of their interactions to develop novel strategies for controlling whitefly infestations. Next-generation sequencing technologies have revolutionized the study of insect–plant interactions. In this study, we employed transcriptome sequencing to investigate the gene expression changes in tobacco plants subjected to different treatments, specifically infestation by the whitefly *B. tabaci*. To gain insight into tobacco gene expression, we constructed cDNA libraries at different time points following whitefly infestation and analyzed the DEGs. Our statistical analysis revealed that the majority of DEGs were concentrated at 24 h after tobacco infestation by *B. tabaci*. Interestingly, we observed a higher number of downregulated genes compared to upregulated genes at this time point, indicating that whitefly infestation predominantly suppresses gene expression in tobacco.

Previous research has demonstrated that plants respond to pathogen exposure by PRRs, leading to downstream defense responses such as MAPK cascades, calcium flux, ROS bursts, transcriptional reprogramming, and phytohormone signaling pathways [29,30,31,32,33]. In our study, we found that the genes upregulated in response to whitefly infestation were associated with ROS and hormonal pathways. Additionally, GO enrichment pathway analysis highlighted their involvement in the response to stimulus, response to biotic stimulus, and innate immune responses. These findings suggest that the infestation of tobacco by *B. tabaci* triggers the activation of the plant innate immune system and responses to external stimuli. Notably, our study revealed significant changes in nearly 50 DEGs related to PRRs following whitefly infestation. Plants utilize PRRs to rapidly detect potential dangers posed by microbes and pests [11]. Based on this observation, we hypothesize that whitefly infestation in tobacco activates plant PRRs, thereby initiating downstream defense responses to counter the infestation.

TFs are key components of plant signaling networks that regulate gene expression in response to various factors, including plant growth, development, and biotic and abiotic stresses [34,35]. In our study, several TFs, including WRKY TFs and ERFs, were identified among the DEGs. Previous research has highlighted the crucial role of WRKY TFs in plant defenses against sap-sucking insects like aphids, planthoppers, and whitefly [36]. For example, WRKY89 enhanced resistance to the white-backed planthopper *Sogatella furcifera* in rice, and altered expression levels of six WRKY genes were observed in cotton following whitefly infestation [37,38]. Additionally, the upregulation of *N. tabacum* WRKY6 was reported when tobacco plants were infested with whiteflies [39]. In our study, with the exception of the downregulated WRKY 26 gene at 24 h, all other WRKY TFs, including WRKY6, were significantly upregulated. ERFs, another group of plant-specific TFs, are involved in regulating plant development and stress responses. Notably, ERF1 has been shown to play a key role in activating plant defenses against necrotrophic pathogens, while OsERF3 in rice regulates resistance to chewing or piercing/sucking insects through metabolic regulation [40,41,42]. Additionally, ERF5 has been implicated in plant innate immunity [43]. In our study, we observed a significant upregulation of ERF1, ERF1a, ERF2b, ERF4, and ERF5. Collectively, these results indicate that TFs, particularly WRKY TFs and ERFs, play a crucial role in regulating gene expression related to plant defense against whitefly infestation.

## 5. Conclusions

In conclusion, our study sheds light on the complex interplay between whiteflies and host plants, providing valuable insights into the gene expression changes in tobacco following *B. tabaci* infestation. The observed activation of the plant immune system, the involvement of PRRs, and the significant upregulation of TFs, especially WRKY TFs and ERFs, underscore their essential roles in plant defense against whitefly infestations. Understanding these molecular mechanisms will contribute to the development of effective strategies for whitefly control and crop protection. Further research should focus on unraveling the intricate details of these interactions and exploring the functional significance of specific genes involved in the plant defense response to whitefly infestations.

## Figures and Tables

**Figure 1 genes-14-01640-f001:**
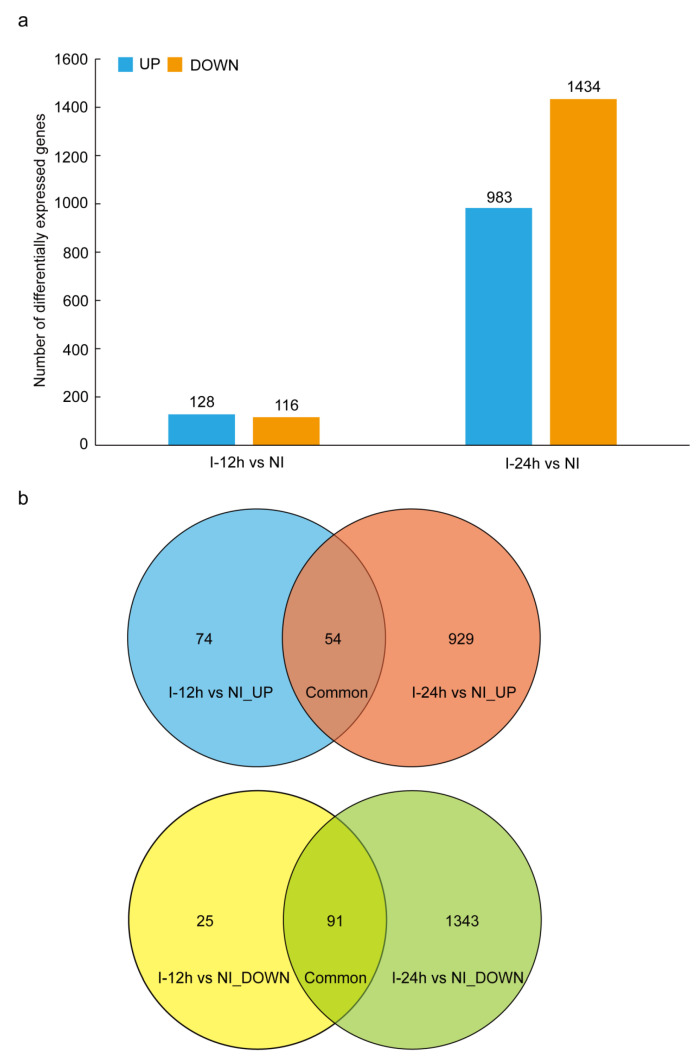
Analysis of differentially expressed genes (DEG) upon *B. tabaci* infestation. (**a**) Number of DEGs in tobacco infested by *B. tabaci* at different time points. (**b**) The Venn diagram shows the number of specific and common DEGs that are upregulated and downregulated at different time points.

**Figure 2 genes-14-01640-f002:**
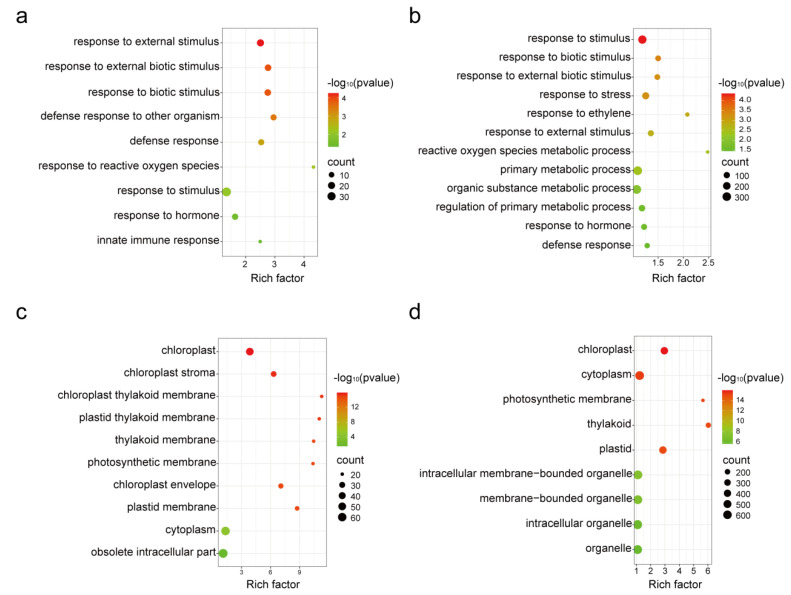
Scatterplot of Gene Ontology (GO) pathway enrichment analysis for differentially expressed genes (DEG). (**a**) Analysis of upregulated DEGs at 12 h post infestation. (**b**) Analysis of upregulated DEGs at 24 h post infestation. (**c**) Analysis of downregulated DEGs at 12 h post infestation. (**d**) Analysis of downregulated DEGs at 24 h post infestation. The rich factor is the ratio of DEGs annotated in a given pathway term to the total number of genes annotated in that pathway term. The gene ratio is represented on the horizontal axis, while the enriched pathway names are shown on the vertical axis. The color scale indicates various *p*-value thresholds, and the size of the dots corresponds to the number of genes associated with each pathway.

**Figure 3 genes-14-01640-f003:**
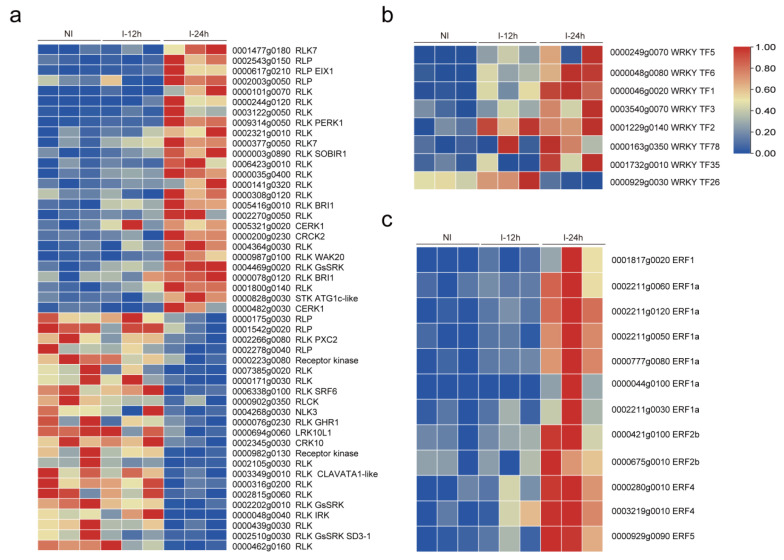
Analysis of differentially expressed genes (DEG) associated with plant resistance. The expression patterns of receptor-like kinases (RLKs) and receptor-like proteins (RLPs) (**a**), WRKY transcription factors (TFs) (**b**), and ethylene-responsive transcription factors (ERFs) (**c**) were displayed. The heatmap was drawn using TBtools software, and the function “scale” was used for the data by rows. The scale method was “zero to one”. A red color indicates high expression, while a blue color indicates low expression.

**Figure 4 genes-14-01640-f004:**
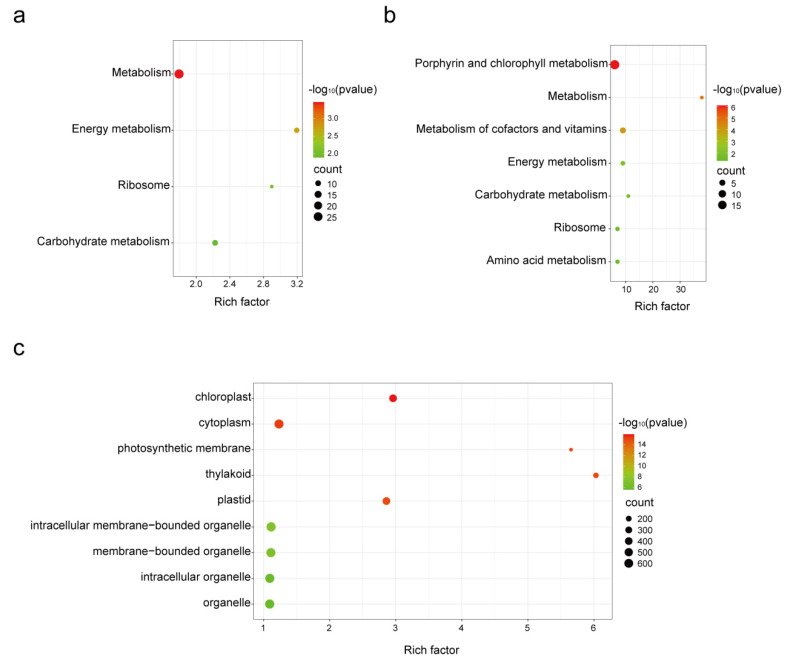
Enrichment analysis of downregulated, differentially expressed genes (DEG) upon *B. tabaci* infestation. (**a**) Analysis of DEGs commonly downregulated at 12 h and 24 h post whitefly infestation. (**b**,**c**) Analysis of DEGs downregulated at 12 h (**b**) and 24 h (**c**) post whitefly infestation.

**Figure 5 genes-14-01640-f005:**
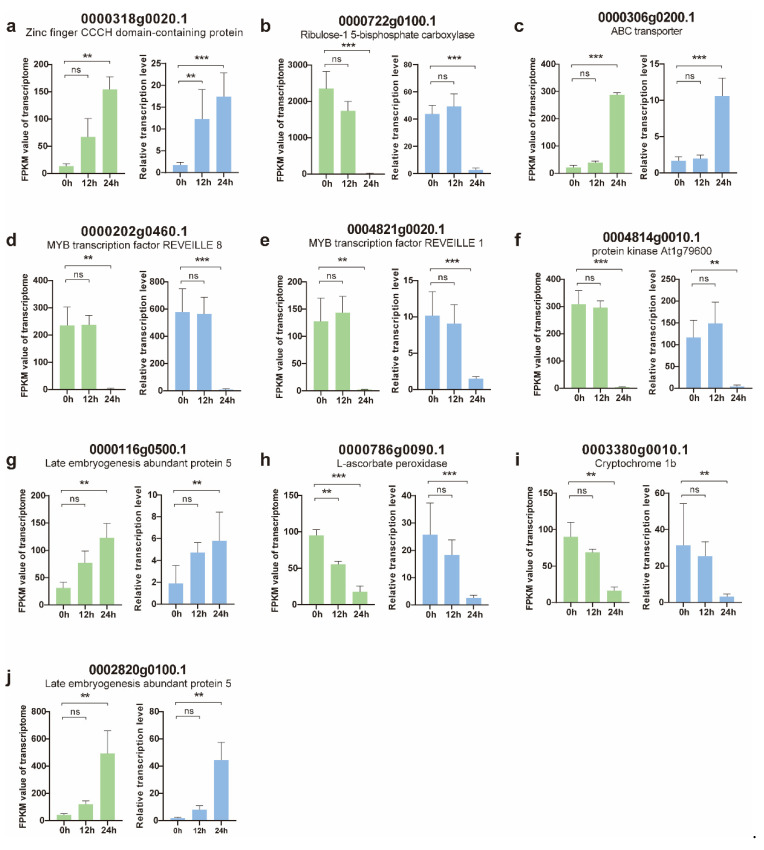
Correlation between transcriptomic data (green) and qRT-PCR results (blue). The relative expression level of each gene was determined using qRT-PCR and was compared with the expression of the transcriptomic data. *p*-values in transcriptomic data were determined with DESeq software, while *p*-values in qPCR were determined with a two-tailed unpaired Student’s *t* test. ** *p* < 0.01; *** *p* < 0.001; ns, not significant. Data are presented as mean values ± SD. (**a**) Zinc finger CCCH domain-containing protein; (**b**) Ribulose-1,5-bisphosphate carboxylase; (**c**) ABC transporter; (**d**) MYB transcription factor REVEILLE 8; (**e**) MYB transcription factor REVEILLE 1; (**f**) Protein kinase At1g79600; (**g**) Late embryogenesis abundant protein 5; (**h**) L-ascorbate peroxidase; (**i**) Cryptochrome b; (**j**) Late embryogenesis abundant protein 5.

**Table 1 genes-14-01640-t001:** Summary of statistics from Illumina sequencing.

Sample	Raw Reads	Clean Reads	Raw Base (G) ^1^	Clean Base (G) ^2^	Error (%) ^3^	Q20 (%) ^4^	Q30 (%) ^5^	GC Content (%)
NI-1	22825325	22389877	6.85	6.72	0.03	97.32	92.53	41.69
NI-2	26091282	24468330	7.83	7.34	0.03	97.89	93.69	43.02
NI-3	28556897	26626176	8.57	7.99	0.03	97.89	93.76	43.18
I-12 h-1	27446453	25817532	8.23	7.75	0.03	97.97	93.88	43.32
I-12 h-2	26080109	24327843	7.82	7.3	0.03	97.95	93.89	43.31
I-12 h-3	26357914	24481241	7.91	7.34	0.03	97.98	93.98	43.36
I-24 h-1	26781167	24971508	8.03	7.49	0.03	97.86	93.69	42.38
I-24 h-2	23722409	23321513	7.12	7	0.03	97.23	92.28	41.91
I-24 h-3	24041972	22384567	7.21	6.72	0.03	97.91	93.79	41.82

^1^ Raw Base: the total number of bases in the raw data. ^2^ Clean Base: the total number of bases in the clean data. ^3^ Error: error rate of average base sequencing. ^4^ Q2: percentage of bases with Phred quality scores > 20. ^5^ Q30: percentage of bases with Phred quality scores > 30.

## Data Availability

All sequencing data generated in this study were submitted to the National Genomics Data Center under accession number PRJCA018862.

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
