# Peer review of "Transcriptomic Analysis of Tobacco Plants in Response to Whitefly Infection"

_genes, 2023, doi:10.3390/genes14081640_

Round 1

Reviewer 1 Report

This manuscript reported the identification and analysis of the genes responsive to whitefly infestation in tobacco. With RNA seq, a standard and routine protocol has been implemented to characterize the expression patterns of DEGs, followed by qRT-PCR validation for several important TFs and enzymes. This study provides good information for further understanding the molecular mechanism underlying the complex interplay between whiteflies and tobacco hosts. The experiment and data analysis were correctly performed. Conclusion and discussion were reasonably drawn.

The only thing I would like to point out is the real-time PCR validation part.

I wonder how or on what basis the 11 genes were selected. I could not locate any of them in Figure 3.

Figure 5: Instead of using gene ID for each gene, it may be more reader intuitive to use gene name or both.

Good.

Author Response

#Review 1

This manuscript reported the identification and analysis of the genes responsive to whitefly infestation in tobacco. With RNA seq, a standard and routine protocol has been implemented to characterize the expression patterns of DEGs, followed by qRT-PCR validation for several important TFs and enzymes. This study provides good information for further understanding the molecular mechanism underlying the complex interplay between whiteflies and tobacco hosts. The experiment and data analysis were correctly performed. Conclusion and discussion were reasonably drawn.

Response: Thank you for your positive comments.

The only thing I would like to point out is the real-time PCR validation part.

I wonder how or on what basis the 11 genes were selected. I could not locate any of them in Figure 3.

Figure 5: Instead of using gene ID for each gene, it may be more reader intuitive to use gene name or both.

Response: Done. Thank you for pointing out this. In the revised version, we add both gene name and gene ID at top of histogram, which is easier for readers to understand the genes we selected.

   For the criteria of DEGs used in qPCR validation, we select defense-related transcription factors, protein kinase, detoxification proteins, and other proteins that showed significant upregulation/downregulation in transcriptomic analysis. In the revised version, we described the rationality of gene selection as “To confirm the expression of the DEGs, 10 genes that were associated with plant defenses or showed dramatic upregulation/downregulation in the RNA-Seq analysis were selected for qRT-PCR analysis”

Reviewer 2 Report

The authors have analyzed the response of tobacco plants (Nicotiana tabacum) at 12 h and 24 h post whitefly (Bemisia tabaci) infestation using next generation sequencing, search for differentially expressed genes (DEGs), and qRT-PCR to check for quality of some DEGs. The up- and down-regulated genes (mainly transcription factors, receptor-like kinases) at 12 and 24 h were quite different; defense genes were upregulated at both time points; fundamental metabolic and energy processes were suppressed especially at 24 h.

Results shown in Fig. 5 serve to experimentally verify the DEG analysis. Figs 5d and e show MYB TFs, Figs 5g and j show LEA5 proteins; both "pairs" have identical annotations. Are you sure that these "pairs" are different genes and that, if they are different genes, your primers are able to differentiate between these?

Sections 2.1 and 2.2: Plants were cultivated at a temperature of 24±2°C; whiteflies were kept at 27±2°C. At what temperature was performed the infestation? Could this have any influence on the results?

Typos, grammar
==============
12: "tobacco perceive and defense" => "tobacco perceives and defends"

12: "analyze" => "analyzed"

15: "actives" => "activated"

17: "were fiercely" => "were more fiercely"

54: What is "BPH"?

106: "37 ° C" => "37°C"

125, 130, 131, 132, 201: Use mathemathical characters (italics) for variables P, N, M, and m.

144: "5min at 95 °C" => "5 min at 95°C"

Table 1 (or corresponding text lines 152ff): Please, explain shortly the table headers ("RAw Base" vs "Clean base", Effective rate, Q20, Q30, ...).

165: "Differently expression genes" => "Differently expressed genes"

225: "(ERFs) were" => "(ERFs) (c) were"

226: "heatmap." => "heatmap (top right)."
    What are the values represented in the heatmap?

Legend Fig. S1: "results were shown." => "results are shown."

Fig. 5: Add annotations in addition to the gene IDs (for example, in the figure legend); this will help a lot to understand the context of the shown analysis.

319ff: "s1, Figure S1. Principal ... shown. Table S1. Twelve ... validation Table S2. Primers ... study. Table S3. Differentially ... infestation. Table S4. Differentially ... infestation" => "s1: Figure S1: Principal ... shown; Table S1: Twelve ... validation; Table S2: Primers ... study; Table S3: Differentially ... infestation; Table S4: Differentially ... infestation."

323: "Bemisia tabaci" => "Bemisia tabaci" (italics)

Author Response

#Review 2

The authors have analyzed the response of tobacco plants (Nicotiana tabacum) at 12 h and 24 h post whitefly (Bemisia tabaci) infestation using next generation sequencing, search for differentially expressed genes (DEGs), and qRT-PCR to check for quality of some DEGs. The up- and down-regulated genes (mainly transcription factors, receptor-like kinases) at 12 and 24 h were quite different; defense genes were upregulated at both time points; fundamental metabolic and energy processes were suppressed especially at 24 h.
Response: We greatly appreciate your valuable comments and constructive suggestion to our work.

Results shown in Fig. 5 serve to experimentally verify the DEG analysis. Figs 5d and e show MYB TFs, Figs 5g and j show LEA5 proteins; both "pairs" have identical annotations. Are you sure that these "pairs" are different genes and that, if they are different genes, your primers are able to differentiate between these?

Response: Thank you for your suggestion. In the revised version, we carefully analyze the nucleic and amino acid identity of “paired” sequences. The results showed that although two MYB transcription factor exhibit 51.02% (e-value = 4e-28) identity in amino acid level, no identity in nucleic acid level was detected. Similar results were found in two late embryogenesis abundant proteins. Therefore, the sequence-specific primers can easily distinguish these sequence pairs in nucleic acid level. To avoid misunderstanding, in the revised version, we re-blast these sequence against NCBI nr database, and annotate these genes in detail.

Sections 2.1 and 2.2: Plants were cultivated at a temperature of 24±2°C; whiteflies were kept at 27±2°C. At what temperature was performed the infestation? Could this have any influence on the results?
Response: Thank you for pointing out this. The infestation experiment was conducted at 27±2°C. To minus the effect of temperature changes on tobacco, the plants usually transferred from climate chamber of 24±2°C to 27±2°C one day before experiments. Also, all treatments were undergone the same temperature changes. Therefore, the effect of temperature change might be not significant. In the revised version, we addressed that “the tobacco plants were transferred to climate chamber (27 ± 2°C) one day before treatments” in the Methods section.

Typos, grammar
==============
12: "tobacco perceive and defense" => "tobacco perceives and defends"

Response: Done.

12: "analyze" => "analyzed"

Response: Done.

15: "actives" => "activated"

Response: Done.

17: "were fiercely" => "were more fiercely"

Response: Done.

54: What is "BPH"?

Response: Done.

106: "37 ° C" => "37°C"

Response: Done.

125, 130, 131, 132, 201: Use mathemathical characters (italics) for variables P, N, M, and m.

Response: Done.

144: "5min at 95 °C" => "5 min at 95°C"

Response: Done.

Table 1 (or corresponding text lines 152ff): Please, explain shortly the table headers ("RAw Base" vs "Clean base", Effective rate, Q20, Q30, ...).

Response: Done. In the revised version, we provide the detail description of these parameter in the table and the main text.

165: "Differently expression genes" => "Differently expressed genes"

Response: Done.

225: "(ERFs) were" => "(ERFs) (c) were"

Response: Done.

226: "heatmap." => "heatmap (top right)."
    What are the values represented in the heatmap?

Response: Done.

Legend Fig. S1: "results were shown." => "results are shown."
Response: Done.

Fig. 5: Add annotations in addition to the gene IDs (for example, in the figure legend); this will help a lot to understand the context of the shown analysis.
Response: Done.

319ff: "s1, Figure S1. Principal ... shown. Table S1. Twelve ... validation Table S2. Primers ... study. Table S3. Differentially ... infestation. Table S4. Differentially ... infestation" => "s1: Figure S1: Principal ... shown; Table S1: Twelve ... validation; Table S2: Primers ... study; Table S3: Differentially ... infestation; Table S4: Differentially ... infestation."

Response: Done.

323: "Bemisia tabaci" => "Bemisia tabaci" (italics)

Response: Done.

Reviewer 3 Report

This paper investigates the gene expression patterns of tobacco in response to whitefly infestation using an RNA-seq-based approach. The authors observed that whitefly infestation activates plant defense, resulting in a stronger immune response 24 hours post-infestation compared to 12 hours post-infestation. While the topic is novel and the methodology is sound, the findings may lack sufficient novelty. Nevertheless, with more effort, the authors have the potential to uncover more biological insights from their data. Several areas require improvement in explaining the methods clearly. The major comments address crucial issues related to data availability, data processing, and statistical analysis. Additionally, minor comments highlight areas for improvement in figure labeling and clarifying technical details.

Major Comments:

1. The authors should make the RNA-seq data available to the public, preferably by depositing it in databases like Gene Expression Omnibus, to ensure transparency and facilitate further research by the scientific community.

2. Lines 115 – 119: The authors should provide more clarity on the preprocessing of the RNA-seq data. Details on the internal software/scripts used for filtering low-quality reads, the alignment tool used to align reads to the reference genome, and any other steps of RNA-seq data processing should be included. A diagram illustrating the RNA-seq data processing workflow would be beneficial for readers' understanding.

3. Lines 121 – 125: The authors were ambiguous about how they conducted differential expression analysis. While DESeq was mentioned on line 122, Cuffdiff was referred to on line 124. The authors should specify which method they used for differential expression analysis and make the codes available for reproducibility.

4. Lines 124 – 125: The criteria for defining differentially expressed genes should be better clarified. Were p-values adjusted for multiple hypothesis testing? If so, the authors should specify the adjustment method used (e.g., BH method). Details on how p-values were adjusted for pathway enrichment analysis should also be provided.

5. Lines 156 – 157: The authors mentioned that quality control of raw reads was conducted using FastQC. The method of quality control should be included in the methods section (end of section 2.3) to provide a comprehensive description of the quality control process.

6. The major finding that whitefly infestation suppresses plant metabolism and gene expression requires further clarification. Which metabolic pathways are downregulated? Are there specific metabolic pathways upregulated by B. tabaci feeding? Have the authors observed any genes that are differentially expressed only at 12 hours post infestation? Some of these genes may resemble “immediate early” genes, playing critical roles in gene regulation in pest-infested plants.

Minor Comments:

1. Figure 5: Genes were labeled using database accession codes. It would be more reader-friendly to label these transcripts with gene symbols and gene descriptions for better comprehension.

2. Lines 123 – 124: The correct abbreviation for "FPKM" is "fragments," not "reads," per kilobase of transcript per million mapped reads. The authors should clarify whether they used FPKM or RPKM for their analysis.

Overall, this paper presents a novel study investigating the gene expression patterns of tobacco in response to whitefly infestation. While the methodology is sound, the authors should address the major comments regarding data availability, data processing, and statistical analysis to enhance the study's quality. Additionally, minor improvements in figure labeling and technical details would further improve the paper's clarity. The topic has potential, and with more effort, the authors can uncover more biologically significant insights from their data. Thus, I recommend a revision of the manuscript before considering it for publication.

This manuscript contains several grammar mistakes that need to be addressed. For example, within the abstract alone, more than five grammar mistakes were detected. The authors should carefully proofread their manuscript to correct these errors. Some specific corrections include:

Line 12: "how tobacco perceive and defense against whitefly" should be "how tobacco perceive and defend themselves against whiteflies."

Line 15: "whitefly infestation actives plant defense" should be "whitefly infestation activates plant defense."

Line 17: "noteworthily" should be "notably."

Line 18: "that of 12 h" should be "those of 12 h."

Line 18: "an increased immunity along with whitefly infestation" should be "an increased immunity induced by whitefly infestation."

Line 20: "our study provides a comprehensive insight" should be "our study provides comprehensive insights."

Line 21: ", and provides" should be omitted to avoid redundancy.

Addressing these grammar mistakes will enhance the readability and clarity of the paper.

Author Response

#Review 3

This paper investigates the gene expression patterns of tobacco in response to whitefly infestation using an RNA-seq-based approach. The authors observed that whitefly infestation activates plant defense, resulting in a stronger immune response 24 hours post-infestation compared to 12 hours post-infestation. While the topic is novel and the methodology is sound, the findings may lack sufficient novelty. Nevertheless, with more effort, the authors have the potential to uncover more biological insights from their data. Several areas require improvement in explaining the methods clearly. The major comments address crucial issues related to data availability, data processing, and statistical analysis. Additionally, minor comments highlight areas for improvement in figure labeling and clarifying technical details.

Response: We greatly appreciate your valuable comments and constructive suggestion to our work. We have revised the manuscript according to your suggestions, which greatly improve the quality of manuscript.

Major Comments:

  1. The authors should make the RNA-seq data available to the public, preferably by depositing it in databases like Gene Expression Omnibus, to ensure transparency and facilitate further research by the scientific community.

Response: Done. Thank you for your suggestion. In the revised period, we submitted the RNA-seq data to the National Genomics Data Center under accession number: PRJCA018862. All readers can access to these data.

  1. Lines 115 – 119: The authors should provide more clarity on the preprocessing of the RNA-seq data. Details on the internal software/scripts used for filtering low-quality reads, the alignment tool used to align reads to the reference genome, and any other steps of RNA-seq data processing should be included. A diagram illustrating the RNA-seq data processing workflow would be beneficial for readers' understanding.

Response: Done. Thank you for your suggestion. In the revised period, we consult the Novogene Company to provide the detail procedure used in preprocessing of the RNA-seq data. The preprocessing of RNA-seq data in this study is regular and widely used in RNA-seq. In the revised version, we provide the detail software used in each of procedure and cite the corresponding references as follow: “The clean reads were obtained using internal software Fastp version 0.23.1 [18] by removing low-quality reads that contained adapters, empty reads or reads with unknown sequences “N” from the raw data. Subsequently, the clean reads from each cDNA library were aligned to the reference genome sequences of N. tabacum (https://solgenomics.net/ftp/genomes/Nicotiana_tabacum/edwards_et_al_2017/assembly/) using Hierarchical Indexing for Spliced Alignment of Transcripts (HISAT2) [19]. The low-quality alignments were filtered with Sequence Alignment/Map tools (SAMtools) [20].”

  1. Lines 121 – 125: The authors were ambiguous about how they conducted differential expression analysis. While DESeq was mentioned on line 122, Cuffdiff was referred to on line 124. The authors should specify which method they used for differential expression analysis and make the codes available for reproducibility.

Response: Done. Thank you for pointing out this and sorry for making mistake. In the revised period, we consult the Novogene Company, which help perform DEG analysis in this study, to validate the DEG analysis procedure. The methods used in this section was rewrite as “Transcripts per million (TPM) expression values were calculated using cufflink [21]. The DESeq2 (http://bioconductor.org/packages/release/bioc/html/DESeq2) was used for identify the differentially expressed genes with default parameters”.

  1. Lines 124 – 125: The criteria for defining differentially expressed genes should be better clarified. Were p-values adjusted for multiple hypothesis testing? If so, the authors should specify the adjustment method used (e.g., BH method). Details on how p-values were adjusted for pathway enrichment analysis should also be provided.

Response: Done. Thank you for pointing out this. The P-values of differentially expressed genes were generated using DESeq2 with default parameters. No further adjustment was performed in this study. The readers can easily reproduce our results using this software.

For the pathway enrichment analysis, we perform it using TBtools software v1.0697. This software was widely used in basic bioinformatics analysis, and we did not need to input specific parameters in pathway enrichment. In the protocol of this software, the one-sided hypergeometric test was addressed in calculating the P-value. However, we did not find the instrument for adjust treatment following the p-value calculation. Overall, the enrichment analysis can be reproduced using TBtools software v1.0697.

  1. Lines 156 – 157: The authors mentioned that quality control of raw reads was conducted using FastQC. The method of quality control should be included in the methods section (end of section 2.3) to provide a comprehensive description of the quality control process.

 Response: Done. Thank you for your suggestion. In the revised version, we provide the method in assess the quality control of raw reads as “Quality control of reads was assessed using FastQC program (version 0.11.3) [19]. It performed a series of analysis modules on raw data and created a report with statistics for the data analyzed, including error rate of average base sequencing, percentage of bases with Phred quality scores>20 (Q20) or >30 (Q30)”.

  1. The major finding that whitefly infestation suppresses plant metabolism and gene expression requires further clarification. Which metabolic pathways are downregulated? Are there specific metabolic pathways upregulated by B. tabaci feeding? Have the authors observed any genes that are differentially expressed only at 12 hours post infestation? Some of these genes may resemble “immediate early” genes, playing critical roles in gene regulation in pest-infested plants.

Response: Done. Thank you for your suggestion. In the revised version, we addressed that “genes associated with energy metabolism, carbohydrate metabolism, ribosomes, and photosynthesis are suppressed”. For upregulated genes upon B. tabaci feeding, we found that genes associated with reactive oxygen species metabolism, organic substance metabolism, and regulation of primary metabolism were significantly enriched. In addition, we also investigate the genes that specifically differentially expressed at 12 hours post infestation. The results showed that a total of 89 DEGs specifically differentially expressed at 12 h, but not 24 h, with the majority of them specifically upregulated. In the revised version, we described that “There were 89 DEGs specifically differentially expressed at 12 h, but not 24 h, when compared with the non-infested control. Notably, the majority of these genes (68, 76.4%) were specifically upregulated at the early stage, including ABC transporters, serine threonine tyrosine kinase, calmodulin-binding protein, BHLH transcription factor, and UDP-glucosyltransferase. These genes potentially act as immediate-early expressed genes [26] that efficiently response to whitefly infestation.”

Minor Comments:

  1. Figure 5: Genes were labeled using database accession codes. It would be more reader-friendly to label these transcripts with gene symbols and gene descriptions for better comprehension.

Response: Done. Thank you for pointing out this. In the revised version, we add both gene name and gene ID at top of histogram, which is easier for readers to understand the genes we selected.

  1. Lines 123 – 124: The correct abbreviation for "FPKM" is "fragments," not "reads," per kilobase of transcript per million mapped reads. The authors should clarify whether they used FPKM or RPKM for their analysis.

Response: Done. Thank you for pointing out this. In the revised version, we check the methods used in DEG analysis with the Novogene Company, which help perform this experiment, and confirmed that we use the transcripts per million (TPM). Therefore, we correct the description, and provide the detail methods in calculating TPM. Sorry about this.

Overall, this paper presents a novel study investigating the gene expression patterns of tobacco in response to whitefly infestation. While the methodology is sound, the authors should address the major comments regarding data availability, data processing, and statistical analysis to enhance the study's quality. Additionally, minor improvements in figure labeling and technical details would further improve the paper's clarity. The topic has potential, and with more effort, the authors can uncover more biologically significant insights from their data. Thus, I recommend a revision of the manuscript before considering it for publication.

Response: Thank you for your positive comments. We have revised the manuscript according to your suggestion.

Comments on the Quality of English Language

This manuscript contains several grammar mistakes that need to be addressed. For example, within the abstract alone, more than five grammar mistakes were detected. The authors should carefully proofread their manuscript to correct these errors. Some specific corrections include:

Response: Thank you for pointing out. In the revised version, we consult an English Language Service to improve our grammar. We hope our manuscript meet the journal well.

Line 12: "how tobacco perceive and defense against whitefly" should be "how tobacco perceive and defend themselves against whiteflies."

Response: Done.

Line 15: "whitefly infestation actives plant defense" should be "whitefly infestation activates plant defense."

Response: Done.

Line 17: "noteworthily" should be "notably."

Response: Done.

Line 18: "that of 12 h" should be "those of 12 h."

Response: Done.

Line 18: "an increased immunity along with whitefly infestation" should be "an increased immunity induced by whitefly infestation."

Response: Done.

Line 20: "our study provides a comprehensive insight" should be "our study provides comprehensive insights."

Response: Done.

Line 21: ", and provides" should be omitted to avoid redundancy.

Response: Done.

Addressing these grammar mistakes will enhance the readability and clarity of the paper.

 Response: Thank you for pointing out. The English has been modified. Thank you again for your suggestion.

Round 2

Reviewer 3 Report

The revised version of the manuscript has been meticulously modified in response to the previous comments. I am pleased to note that the authors have diligently addressed all the major and minor concerns I raised. The paper now demonstrates a commendable level of clarity and organization. I have carefully re-evaluated the manuscript and find the revisions to be satisfactory. I am confident that the study has been significantly improved and is now suitable for publication.